# Joint Constraints Based Dynamic Calibration of IMU Position on Lower Limbs in IMU-MoCap

**DOI:** 10.3390/s21217161

**Published:** 2021-10-28

**Authors:** Qian Hu, Lingfeng Liu, Feng Mei, Changxuan Yang

**Affiliations:** School of Information Engineering, East China Jiao Tong University, Shuanggang Dong Dajie 808, Nanchang 330013, China; huqian_ecjtu@163.com (Q.H.); mei_feng3@163.com (F.M.); changxuan_y@163.com (C.Y.)

**Keywords:** IMU, MoCap, DWPSO, GWO, Gauss–Newton, joint constraints

## Abstract

The position calibration of inertial measurement units (IMUs) is an important part of human motion capture, especially in wearable systems. In realistic applications, static calibration is quickly invalid during the motions for IMUs loosely mounted on the body. In this paper, we propose a dynamic position calibration algorithm for IMUs mounted on the waist, upper leg, lower leg, and foot based on joint constraints. To solve the problem of IMUs’ position displacement, we introduce the Gauss–Newton (GN) method based on the Jacobian matrix, the dynamic weight particle swarm optimization (DWPSO), and the grey wolf optimizer (GWO) to realize IMUs’ position calibration. Furthermore, we establish the coordinate system of human lower limbs to estimate each joint angle and use the fusion algorithm in the field of quaternions to improve the attitude calibration performance of a single IMU. The performances of these three algorithms are analyzed and evaluated by gait tests on the human body and comparisons with a high-precision IMU-Mocap reference device. The simulation results show that the three algorithms can effectively calibrate the IMU’s position for human lower limbs. Additionally, when the degree of freedom (DOF) of a certain dimension is limited, the performances of the DWPSO and GWO may be better than GN, when the joint changes sufficiently, the performances of the three are close. The results confirm that the dynamic calibration algorithm based on joint constraints can effectively reduce the position offset errors of IMUs on upper or lower limbs in practical applications.

## 1. Introduction

In recent years, inertial measurement units (IMUs) have attracted increasing interest in the field of human motion analysis. The wearable sensor motion capture system is less costly, more flexible, and more portable than optical camera-based motion capture devices [1,2]. By mounting IMUs on each limb of human bodies, the real-time tracking and motion data analysis of human postures can be realized. The IMU-based motion capture and analysis have shown substantial applications in athletic training, e.g., golf training, baseball training, dart-throwing training, etc. [3,4,5,6]. It also has promising application prospects in medical rehabilitation training. In [7], the IMU-Mocap system is applied to determine the level of autonomy for patients with Parkinsonism syndromes. To obtain the information of body motions, the installation position of the IMUs and the variation of joint space position should be accurately measured. The works [1,8] analyzed the influence of IMUs’ positions or directions on the accuracy of motion evaluation, and it will further affect the variation of the joint angle. However, in practical applications, different types of clothing materials and muscle stretching during exercise will cause IMU position displacements. A direct consequence of IMU displacement is the difference of the derived joint position relative to the pre-calibration. Therefore, an effective IMU position calibration method is necessary.

Regarding IMU position displacement, there are currently two types of IMU position calibration, divided into static calibration, e.g., quiet standing, and dynamic calibration, e.g., knee flexion. For static calibration, the work [9] realized an IMU-to-body calibration based on preset static postures. In [10,11], the direction of the IMUs was estimated by a T-pose or N-pose to ensure that each frame in the IMUs aligns with the known direction in that posture. For dynamic calibration, the works [12,13] calibrated the IMUs bound to the upper leg and lower leg by using flexion/extension (FE) and abduction/adduction (AA) of the knee joint, so that the angle of the knee joint is equal to zero in the standing posture. The studies [14,15] proposed a simple calibration scheme, which does not need to specify the motion of limbs, and used human gait analysis to align the direction of IMUs to the body. However, the above methods do not estimate the positions of the sensors relative to the adjacent limbs, which is critical information for calculating the joint angle using IMUs, especially during fast rotations of the joint [16], and establish a motion chain model in high-speed motions [17]. The study [18] proposed a position estimation algorithm to estimate the position of IMUs relative to the limbs based on the least-squares optimization. This approach was further extended in [19,20] for gait analysis and the angles of the knee and ankle joint were estimated; it was then applied to the rehabilitation system of human limbs.

The limitations of the above calibration methods can be summarized as follows:

(1) The wearing position of the sensor needs to be fixed or special tools are required [9,10,11].

(2) The limbs are required to perform specific motions, but it is difficult for volunteers with damaged joints to complete them. Even volunteers with normal physical activities need to be guided by professionals [12,13].

(3) The direction of IMUs must be estimated in advance to complete the calibration, and it is easy to make mistakes, especially when using a magnetometer, which is vulnerable to the interference of magnetic field [14,15,21].

(4) When the joint rotates, it cannot be fully rotated in all specified directions, resulting in the decline of calibration accuracy [18,19].

### Problem Statement

Considering the limitations of previous calibration algorithms, and based on the joint constraints proposed in [18], in this work, we aim to study the influence of different algorithms on the dynamic calibration performance of IMUs’ position based on joint constraints. Furthermore, we aim to establish the human lower limb coordinate system and calculate the joint angle to study the influence of IMUs’ position accuracy on human gait space-time parameters. The study [18] did not provide research on the difference of results caused by different joint motion types in the actual motion process. Due to the different variation ranges of the hinge joint and spherical joint, the degree of freedom (DOF) of the joint will change. When the joint rotation is insufficient, the result of the Gaussian–Newton (GN) algorithm based on the Jacobian matrix may be inaccurate. To solve the IMU position displacement and consider the influence of the change of joint DOF on the calibration algorithm, we introduce the dynamic weight particle swarm optimization (DWPSO) [22] and grey wolf optimizer (GWO) [23] to realize the position calibration of IMUs based on the joint constraints, and the calibration results of the two algorithms are compared with GN. The main contents of this work are as follows:

(1) The four IMUs are bound to the waist, upper leg, lower leg, and foot, respectively, for the gait experiment, and the data of accelerometers and gyroscopes of each IMU are collected. High-precision IMU-Mocap equipment is bound on the lower limbs for synchronous data acquisition with IMUs. High-precision motion capture equipment is only for reference. In addition, we place IMUs in two different positions, and three subjects are tested in two positions, including one female of height 165 cm, and two males of height 175 cm and 180 cm respectively.

(2) The collected data are substituted into the GN, DWPSO, and GWO for position estimation to obtain the position information of IMUs relative to the limbs.

(3) Using the calibrated IMU position information, we establish the human lower limb coordinate system to calculate the angles of the hip, knee, and ankle joint in each DOF. Additionally, to improve the performance of attitude calibration, the quaternion fusion algorithm is used to fuse the data of the accelerometer and gyroscope of single IMU.

(4) The performance of the three algorithms is evaluated by comparing them with the high-precision IMU-Mocap reference device.

The following of the paper is organized into 5 parts. In Section 2, the IMU position calibration model of the spherical joints and the hinge joints is introduced, respectively. In Section 3, analyzing the performance of the GN, and points out its limitations. Under the same constraints as the GN, the DWPSO and GWO are used to calibrate IMUs’ positions. Section 4 establishes the coordinate system of human lower limbs, combines the position information of IMUs and the attitude of single IMU to calculate the joint angle of human lower limbs during walking. Section 5 introduces the experimental test device and test scheme and analyzes the test data by analyzing the angle variation of each joint angle to verify the performance of the three calibration algorithms. Finally, Section 6 summarizes the study.

## 2. IMU Position Calibration Principle

In this study, we focus on the calibration of IMUs’ positions relative to the lower limb joints. According to the international society of biomechanics (ISB) standard [24] and the joints coordinate system defined in human anatomy [25], 3D rotation for the lower limbs joints can be defined as: (1) hip: flexion/extension ($H_{\mathrm{FE}}$), abduction/adduction ($H_{\mathrm{AA}}$), internal/external rotation ($H_{\mathrm{IE}}$), (2) knee: flexion/ extension ($K_{\mathrm{FE}}$), abduction/adduction ($K_{\mathrm{AA}}$), internal/external rotation ($K_{\mathrm{IE}}$), (3) ankle: flexion/extension ($A_{\mathrm{FE}}$), abduction/adduction ($A_{\mathrm{AA}}$), internal/external rotation ($A_{\mathrm{IE}}$).

For the IMU position estimation, human lower limbs can be simplified as rigid segments connected by joints. Figure 1 presents a model of human left lower limbs; as the human body is symmetrical, it can also be applied to the right lower limbs. The four IMUs are denoted as *S*, S∈{A,B,C,D} being mounted on the waist, upper leg, lower leg, and foot. The hip, knee, and ankle joints are denoted as $J_{i}$, $J_{i}\in \left\{J_{H},\;J_{K},\;J_{A}\right\}$. $r_{J_{K}}$ is the rotation axis of the knee joint. $O_{g}$ is the global coordinate system, which represents the coordinate system of the 3D space object. $O_{s}$, S∈{A,B,C,D} is the sensor coordinate system, which takes IMUs’ center as the coordinate origin.

The human lower limb joints can be classified into a spherical joint and hinge joint, where the hip and ankle are spherical joints and the knee joint is the hinge joint. At time step *t* (t=1…n), the accelerations measured by the accelerometers are denoted as $a_{S}\left(t\right)$. The angular velocities measured by the gyroscope are denoted as $w_{S}\left(t\right)$, S∈{A,B,C,D}.

(1) The spherical joint

The spherical joint is analyzed on the hip joint as an example and is applicable for the ankle joint. Assuming A and B are connected through the joint $J_{H}$, the position of the two IMUs relative to the joint can be determined from the sequence of measurements of A and B. Let $V_{J_{H},\;S}={\left[x_{J_{H},\;S},\;y_{J_{H},\;S},\;z_{J_{H},\;S}\right]}^{T}$, S∈{A,B} denote the vector pointing from the joints center to the origin of the two IMUs coordinate systems in Figure 1. Through the IMU position estimation algorithm proposed in [18], the spherical joint model is defined by Equation (Equation 1).
(1)$$∥a_{A}\left(t\right)-\Gamma_{A}\left(t\right)∥-∥a_{B}\left(t\right)-\Gamma_{B}\left(t\right)∥=0,$$
(2)$$\Gamma_{S}\left(t\right)=w_{S}\left(t\right)\times \left(w_{S}\left(t\right)\times V_{J_{H},\;S}\right)+\alpha_{S}\left(t\right)\times V_{J_{H},\;S},S\in \left\{A,B\right\},$$
where $\left|\left|\cdot \right|\right|$ is the norm for vectors, × is a cross product. The angular acceleration calculated by the angular velocity is denoted as $\alpha_{S}$, which is defined by [19].

The IMU position displacement will cause the errors of Equation (Equation 1), and the equation is not equal on the left and right. The errors are defined by Equation (Equation 3).
(3)$$e_{J_{H}}\left(t\right)=∥a_{A}\left(t\right)-\Gamma_{A}\left(t\right)∥-∥a_{B}\left(t\right)-\Gamma_{B}\left(t\right)∥$$

(2) The hinge joint

The knee joint is a hinge joint, and the model of the knee joint rotation axis estimation proposed in [18] is defined by Equation (Equation 4).
(4)$$∥w_{B}\left(t\right)\times r_{J_{K},\;B}∥-∥w_{C}\left(t\right)\times r_{J_{K},\;C}∥=0,$$
where $r_{J_{K},\;B}$, $r_{J_{K},\;C}$ are the coordinates of the unit vector parallel to the knee joint axis in the $O_{B}$ and $O_{C}$.

Figure 2 show the coordinates of $r_{J_{K},\;B}$ and $r_{J_{K},\;C}$ in spherical coordinates, and converts it to the rectangular coordinate system in Equation (Equation 5)
(5)$$r_{J_{K},\;S}=\left[\cos\left(\varphi_{S}\right)\cos\left(\theta_{S}\right),\;\cos\left(\varphi_{S}\right)\sin\left(\theta_{S}\right),\;\sin\left(\theta_{S}\right)\right],S\in \left\{B,C\right\},$$
where $\varphi \in \left[0,\;\pi \right]$ is the pitch angle, $\theta \in \left[0,\;2\pi \right]$ is the yaw angle. The IMU position displacement will cause Equation (Equation 4) to be unequal on the left and right. The errors are defined by Equation (Equation 6).
(6)$$e_{J_{K}}\left(t\right)=∥w_{B}\left(t\right)\times r_{J_{K},\;B}∥-∥w_{C}\left(t\right)\times r_{J_{K},\;C}∥$$

## 3. Calibration Algorithm Design

### 3.1. Gauss–Newton Method for IMUs Position Calibration

By the analysis of joint constraints in Section 2, we use the Gaussian–Newton (GN) algorithm based on the Jacobian matrix to calculate Equations (Equation 3) and (Equation 6).

For Equation (Equation 3), the optimization problem is expressed by Equation (Equation 7).
(7)$$\begin{array}{cc} \; & \min_{x_{J_{H}}}\sum_{t=1}^{n}e_{J_{H}}^{2}\left(t\right),\\ x_{J_{H}} & ={\left[V_{J_{H},\;A},\;V_{J_{H},\;B}\right]}^{T},\\ e_{J_{H}}\left(t\right) & =∥a_{A}\left(t\right)-\Gamma_{A}\left(t\right)∥-∥a_{B}\left(t\right)-\Gamma_{B}\left(t\right)∥, \end{array}$$
where $x_{J_{H}}$ is the vector containing IMUs’ position parameters, and $x_{J_{H},\;S},\;y_{J_{H},\;S},\;z_{J_{H},\;S}$, S∈{A,B} are in the range [−0.2,0.2].

The iteration steps at time t are described as follows:

(1) Randomly generate initial values of $x_{J_{H}}^{\gamma }$, γ is the number of iterations.

(2) Calculate the deviation vector $e_{J_{H}}$ using Equation (Equation 7).

(3) Calculate the Jacobian matrix $J=\frac{de_{J_{H}}}{dx_{J_{H}}}$ using Equation (Equation 8), and then calculate the generalized inverse matrix of J, which is pinV(J).
(8)$$J=\left[\begin{array}{cc} \frac{\partial e_{J_{H}}\left(1\right)}{\partial V_{J_{H},\;A}} & \frac{\partial e_{J_{H}}\left(1\right)}{\partial V_{J_{H},\;B}}\\ \vdots & \vdots \\ \frac{\partial e_{J_{H}}\left(n\right)}{\partial V_{J_{H},\;A}} & \frac{\partial e_{J_{H}}\left(n\right)}{\partial V_{J_{H},\;B}} \end{array}\right],$$
where
(9)$$\frac{\partial e_{J_{H}}}{\partial V_{J_{H},\;S}}=-\frac{{\left(a_{S}-\Gamma_{S}\right)}^{T}}{∥a_{S}-\Gamma_{S}∥}\left({\left[w_{S}\right]}_{\times }{\left[w_{S}\right]}_{\times }+{\left[\alpha_{S}\right]}_{\times }\right),S\in \left\{A,\;B\right\}$$
the following symbols are introduced by Equation (Equation 10)
(10)$$\begin{array}{c} {\left[w_{S}\right]}_{\times }=\left[\begin{array}{ccc} 0 & -w_{z} & w_{y}\\ w_{z} & 0 & -w_{x}\\ -w_{y} & w_{x} & 0 \end{array}\right],\\ {\left[\alpha_{S}\right]}_{\times }=\left[\begin{array}{ccc} 0 & -\alpha_{z} & \alpha_{y}\\ \alpha_{z} & 0 & -\alpha_{x}\\ -\alpha_{y} & \alpha_{x} & 0 \end{array}\right], \end{array}$$
where $w_{S}={\left[w_{x},\;w_{y},\;w_{z}\right]}^{T}$, $\alpha_{S}={\left[\alpha_{x},\;\alpha_{y},\;\alpha_{z}\right]}^{T}$.

(4) Update $x_{J_{H}}^{\gamma }$ by Equation (Equation 11) and return to (2).
(11)$$x_{J_{H}}^{\gamma +1}=x_{J_{H}}^{\gamma }-pinv\left(J\right)e_{J_{H}},$$

For Equation (Equation 6), the optimization iteration is expressed by Equation (Equation 12).
(12)$$\begin{array}{cc} \; & \min_{x_{J_{K}}}\sum_{t=1}^{n}e_{J_{K}}^{2}\left(t\right),\\ x_{J_{K}} & ={\left[\varphi_{B},\;\theta_{B},\;\varphi_{C},\;\theta_{C}\right]}^{T},\\ e_{J_{K}}\left(t\right) & =∥w_{B}\left(t\right)\times r_{J_{K},\;B}∥-∥w_{C}\left(t\right)\times r_{J_{K},\;C}∥, \end{array}$$
where $x_{J_{K}}$ is the vector containing knee joint axis position parameters. The iteration steps at time t are described as follows:

(1) Randomly generate initial values of $x_{J_{K}}^{\gamma }$.

(2) Calculate $r_{J_{K},\;S}$ using Equation (Equation 5)

(3) Calculate the deviation vector $e_{J_{K}}$ using Equation (Equation 12).

(4) Calculate the Jacobian matrix $J=\frac{de_{J_{K}}}{dx_{J_{K}}}$ using Equation (Equation 13) and calculate the generalized inverse matrix of J is pinv(J).
(13)$$J=\left[\begin{array}{cc} \frac{\partial e_{J_{K}}\left(1\right)}{\partial r_{J_{K},\;B}} & \frac{\partial e_{J_{K}}\left(1\right)}{\partial r_{J_{K},\;C}}\\ \vdots & \vdots \\ \frac{\partial e_{J_{K}}\left(n\right)}{\partial r_{J_{K},\;B}} & \frac{\partial e_{J_{K}}\left(n\right)}{\partial r_{J_{K},\;C}} \end{array}\right],$$
where
(14)$$\frac{\partial (∥w_{S}\times r_{J_{H},\;S}∥)}{\partial r_{J_{H},\;S}}=\frac{\left(w_{S}\times r_{J_{H},\;S}\right)\times w_{S}}{∥w_{S}\times r_{J_{H},\;S}∥},\;S\in \left\{B,\;C\right\}$$

(5) Update $x_{J_{K}}^{\gamma }$ using Equation (Equation 15) and return to (2).
(15)$$x_{J_{K}}^{\gamma +1}\left(t\right)=x_{J_{K}}^{\gamma }\left(t\right)-pinv\left(J\right)e_{J_{K}}\left(t\right)$$

According to the definition of the DH coordinate system in [26], the three DOF (3-DOF) joints of the hip and ankle can be divided into three hinge joints. Therefore, the position of the IMUs relative to the knee joint can be calculated using the spherical joint approach. The positions of B and C relative to the knee joint can be obtained by Equation (Equation 16).
(16)$$\begin{array}{c} V_{J_{K},\;B}={\tilde{V}}_{J_{K},\;B}-\frac{1}{2}\left(r_{J_{K},\;B}^{T}\cdot {\tilde{V}}_{J_{K},\;B}+r_{J_{K},\;C}^{T}\cdot {\tilde{V}}_{J_{K},\;C}\right)r_{J_{K},\;B},\\ V_{J_{K},\;C}={\tilde{V}}_{J_{K},\;C}-\frac{1}{2}\left(r_{J_{K},\;B}^{T}\cdot {\tilde{V}}_{J_{K},\;B}+r_{J_{K},\;C}^{T}\cdot {\tilde{V}}_{J_{K},\;C}\right)r_{J_{K},\;C}, \end{array}$$
where ${\tilde{V}}_{J_{K},\;B}$ and ${\tilde{V}}_{J_{K},\;C}$ are the estimated by Equation (Equation 7).

By analyzing the algorithm, the limitations of the GN are as follows:

(1) In the process of using the GN, the Jacobi matrix theoretically needs to be positive definite; however, the calculation may not be of full rank. When people walk, the motion of the knee joint is mainly flexion and extension, i.e., there is a significant change in only one DOF. When in other DOF, such as internal/external rotation of the knee joint, $w_{x}=w_{y}=0$ cause $\alpha_{x}=\alpha_{y}=0$. According to the analysis of Equations (Equation 8)–(Equation 10), this will reduce the rank of the Jacobian matrix. For the hip or ankle joint, it is also not guaranteed that each motion produces rotation in all 3-DOF at the same time, which reduces the rank of J. The matrix J may be singular and leads to non-convergence of the algorithm.

(2) In the process of practical calculation, the GN is iterated along with one or two of the matrix entries, which causes the high complexity of the J. Therefore, each joint can only be calculated separately. If the motion data of the whole lower limb is processed at the same time, it will increase the complexity of the algorithm and make the iteration time longer and the performance of the GN will be compromised.

### 3.2. Dynamic Weight Particle Swarm Optimization for IMUs Position Calibration

Both the population algorithm and genetic algorithm simulate the adaptability of the individual population on the basis of natural characteristics and use some transformation rules to solve the optimal solution through the search space. However, individual variation will occur in the process of the genetic algorithm, which cannot completely solve the constraints in the optimization problem [27]. Therefore, we chose the population optimization algorithm to realize the position calibration of IMUs.

Under the same constraints as [18], we introduce dynamic weight particle swarm optimization (DWPSO) to calibrate the position of IMUs, where the dynamic weight is added to the traditional PSO. Unlike the GN, the DWPSO does not need to consider the complexity of the Jacobian matrix calculation.

At time t, let *N* is the number of particles in the population, and ε is the ε-th particle of all particles. The parameter vector containing the IMUs’ positions is expressed by Equation (Equation 17). The vector containing knee joint axis position parameters is expressed by Equation (Equation 18).
(17)$$x_{J_{H},\;\varepsilon }={\left[V_{J_{H},\;A,\;\varepsilon },\;V_{J_{H},\;B,\;\varepsilon }\right]}^{T}\;,$$
(18)$$x_{J_{K},\;\varepsilon }={\left[\varphi_{B,\;\varepsilon },\;\theta_{B,\;\varepsilon },\;\varphi_{C,\;\varepsilon },\;\theta_{C,\;\varepsilon }\right]}^{T},$$

For Equations (Equation 3) and (Equation 6), the optimization is expressed by Equations (Equation 19)–(Equation 21).
(19)$$\eta_{\varepsilon }=\mathrm{argmin}\left(e_{J_{i}}\left(x_{J_{i},\;\varepsilon }^{\gamma }\right)\right),$$
where $\eta_{\varepsilon }$ is individual extreme as optimal value of $x_{Ji}$ found by the particle ε in the iteration, $J_{i}\in \left\{J_{H},\;J_{K}\right\}$. γ is the number of iterations.
(20)$$p=\mathrm{argmin}(e_{J_{i}}\left(\eta_{\varepsilon }\right)),$$
where p is the global extremum, i.e., all particles find the optimal value of $x_{Ji}$ in the iterative.
(21)$$\begin{array}{cc} F_{J_{H},\;\varepsilon } & =\left[\begin{array}{cc} Fx_{J_{H},\;A,\;\varepsilon } & Fx_{J_{H},\;B,\;\varepsilon }\\ Fy_{J_{H},\;A,\;\varepsilon } & Fy_{J_{H},\;B,\;\varepsilon }\\ Fz_{J_{H},\;A,\;\varepsilon } & Fz_{J_{H},\;B,\;\varepsilon } \end{array}\right],\\ F_{J_{K},\;\varepsilon } & ={\left[F\varphi_{B,\;\varepsilon },\;F\theta_{B,\;\varepsilon },\;F\varphi_{C,\;\varepsilon },\;F\theta_{C,\;\varepsilon }\right]}^{T}, \end{array}$$
where $F_{J_{H}}$, $F_{J_{K}}$ are the update speed of particles. The algorithm steps at time t are as follows:

(1) Initialization $x_{Ji,\;\varepsilon }$ and $V_{Ji,\;\varepsilon }$. $x_{J_{H},\;S},\;y_{J_{H},\;S},\;z_{J_{H},\;S}$, S∈{A,B} are following uniform distribution in range [−0.2, 0.2]. $\varphi_{B}$, $\varphi_{C}$ are following uniform distribution in range $\left[0,\;\pi \right]$. $\theta_{B}$, $\theta_{C}$ are following uniform distribution in range $\left[0,\;2\pi \right]$.

(2) Substitute $x_{J_{H},\;\varepsilon }$ into Equation (Equation 3) to obtain $e_{J_{H}}$. $x_{J_{K},\;\varepsilon }$ into Equation (Equation 6) to obtain $e_{J_{K}}$.

(3) According to Equations (Equation 19) and (Equation 20) to calculate $\eta_{\varepsilon }$, p.

(4) Update $x_{Ji,\;\varepsilon }$ and $V_{Ji,\;\varepsilon }$ as follows:(22)$$F_{J_{i},\;\varepsilon }^{\gamma +1}=\mu F_{J_{i},\;\varepsilon }^{\gamma }+\Omega_{1}\partial_{1}\left(\eta_{\varepsilon }-x_{J_{i},\;\varepsilon }^{\gamma }\right)+\Omega_{2}\partial_{2}\left(p-x_{J_{i},\;\varepsilon }^{\gamma }\right),$$
(23)$$x_{J_{i},\;\varepsilon }^{\gamma +1}=x_{J_{i},\;\varepsilon }^{\gamma }+F_{J_{i},\;\varepsilon }^{\gamma +1},$$
where $F_{J_{i},\;\varepsilon }^{\gamma +1}$ and $x_{J_{i},\;\varepsilon }^{\gamma +1}$ are the values after the γ-th update. $\partial_{1}$ and $\partial_{2}$ are following uniform distribution in range [0, 1]. μ is the inertia factor. $\Omega_{1}$ and $\Omega_{2}$ are acceleration constants. This work takes $\Omega_{1}=2$, $\Omega_{2}=2$.

(5) Repeat steps 2∼4 until $e_{Ji}$ convergence.

In step 4 of the algorithm, μ will affect the performance of the algorithm. μ is too large or too small will affect the convergence of the error. Therefore, we introduce a PSO based on dynamic weight. μ decreases exponentially in range $\left[\mu_{\min},\;\mu_{\max}\right]$ as γ increases. Additionally, to avoid falling into a local optimum when μ decreases, random jumps are introduced by Equation (Equation 24).
(24)$$\tilde{\mu }=\mu_{\max}{\left(\frac{\mu_{\max}}{\mu_{\min}}\right)}^{1+\frac{1}{10\partial \frac{\gamma }{\gamma_{\max}}}},$$
where $\tilde{\mu }$ is dynamic weight. The maximum number of iterations is denoted as $\gamma_{\max}$. Let $\mu_{\min}=0.2$, $\mu_{\max}=0.8$. Figure 3 shows the convergence curve of $e_{J_{H}}$ when using fixed weight and the dynamic weight, and the $e_{J_{H}}$ is defined by Equation (Equation 3). When the *N* is small, the performance of dynamic weight is better than the fixed weight. It is suitable for some scenes that require high timeliness. When the *N* is large, the performance of dynamic weight is similar to fixed weight.

### 3.3. Grey Wolf Optimizer for IMUs Position Calibration

To verify the influence of different population algorithms on IMUs’ position calibration performance, we also introduce the grey wolf optimizer (GWO) [23] for position calibration. The GWO is a new optimization algorithm inspired by the hunting and social hierarchical behavior of gray wolves. It randomly generates a set of solutions to form an initial gray wolf group, and then iteratively selects the best three wolves in the population, similar to the optimal solution in the PSO.

The parameter vector containing the IMUs position is expressed by Equation (Equation 25). The vector containing knee joint axis position parameters is expressed by Equation (Equation 26).
(25)$$x_{J_{H},p}={\left[V_{J_{H},\;A},\;V_{J_{H},\;B}\right]}^{T}\;,$$
(26)$$x_{J_{K},p}={\left[\varphi_{B},\;\theta_{B},\;\varphi_{C},\;\theta_{C}\right]}^{T},$$
where *p* is a different individual in the population and $x_{J_{i},\;p}$, $J_{i}\in \left\{J_{H},\;J_{K}\right\}$ is the corresponding position of the individual.

In the GWO iteration, the position of each gray wolf represents a feasible solution in the solution space. Let the wolf with the best position be $p_{\alpha }$; the second-best is $p_{\beta }$; the third-best is $p_{\delta }$. The remaining wolves are $p_{\varepsilon }$, i.e., the wolf pack individuals other than the best three wolves. In each iteration, the three best gray wolves in the current population are retained, and then the positions of other search agents are updated according to their position information. $p_{\alpha },\;p_{\beta },\;p_{\delta }$ and $p_{\varepsilon }$ are constantly updated and iterated until the optimal solution is found. The process of finding the optimal solution is the process of a gray wolf hunting prey.

(1) Encircling Prey
(27)$$\begin{array}{cc} f_{p_{\alpha }} & =h_{1}\cdot x_{J_{i},\;p_{\alpha }}^{\gamma }-x_{J_{i},\;p_{\varepsilon }}^{\gamma },\\ f_{p_{\beta }} & =h_{2}\cdot x_{J_{i},\;p_{\beta }}^{\gamma }-x_{J_{i},\;p_{\varepsilon }}^{\gamma },\\ f_{p_{\delta }} & =h_{3}\cdot x_{J_{i},\;p_{\delta }}^{\gamma }-x_{J_{i},\;p_{\varepsilon }}^{\gamma },\\ h & =2\cdot r_{1} \end{array}$$
where γ is the number of iterations. The positions of $p_{\alpha },\;p_{\beta },\;p_{\delta }$, $p_{\varepsilon }$ in γ-th iteration are denoted by $x_{J_{i},\;p_{\alpha }}^{\gamma },\;x_{J_{i},\;p_{\beta }}^{\gamma },\;x_{J_{i},\;p_{\delta }}^{\gamma }$, $x_{J_{i},\;p_{\varepsilon }}^{\gamma }$, respectively. The position vectors of $p_{\varepsilon }$ relative to $p_{\alpha },\;p_{\beta },\;p_{\delta }$ are denoted as $f_{p_{\alpha }},\;f_{p_{\beta }},\;f_{p_{\delta }}$. The random vector following uniform distribution in range $\left[0,\;1\right]$ is denoted as $r_{1}$. $h\in \{h_{1},\;h_{2},\;h_{3}\}$ is the random weight and decreases nonlinearly in iterations. From the initial iteration to the final iteration, it provides global search in the decision space. When the algorithm falls into local optimization and it is not easy to jump out, the randomness of h plays an important role in avoiding local optimization. Equation (Equation 27) shows that, after moving, the $p_{\varepsilon }$ will move around the target gray wolves $p_{\alpha },\;p_{\beta },\;p_{\delta }$, and its orientation is determined by the size of each dimension and the h.

(2) Hunting
(28)$$\begin{array}{cc} x_{J_{i},\;p_{1}} & =x_{J_{i},\;p_{\alpha }}^{\gamma }-k_{1}\cdot f_{\;p_{\alpha }},\\ x_{J_{i},\;p_{2}} & =x_{J_{i},\;p_{\beta }}^{\gamma }-k_{2}\cdot f_{\;p_{\beta }},\\ x_{J_{i},\;p_{3}} & =x_{J_{i},\;p_{\delta }}^{\gamma }-k_{3}\cdot f_{\;p_{\delta }},\\ k & =2\cdot \psi \cdot r_{2}-\psi , \end{array}$$
(29)$$\begin{array}{cc} x_{J_{i},\;p_{\varepsilon }}^{\gamma +1} & =\frac{x_{J_{i},\;p_{1}}+x_{J_{i},\;p_{2}}+x_{J_{i},\;p_{3}}}{3}, \end{array}$$
where $k\in \{k_{1},\;k_{2},\;k_{3}\}$ is the random weight. The random vector following uniform distribution in range $\left[0,\;1\right]$ is denoted as $r_{2}$. ψ is the convergence factor, which decreases linearly from 2 to 0 with the number of iterations. Combining Equations (Equation 27) and (Equation 28) shows that the $p_{\varepsilon }$ moves its position by observing the positions of $p_{\alpha },\;p_{\beta },\;p_{\delta }$, and denoted as $x_{J_{i},\;p_{1}},\;x_{J_{i},\;p_{2}},\;x_{J_{i},\;p_{3}}$, respectively. Then, use Equation (Equation 29) to determine the moving direction of the prey and update its position, i.e., $x_{J_{i},\;p_{\varepsilon }}^{\gamma +1}$ is the updated position of the $p_{\varepsilon }$. Through a continuous iterative search, the optimal solution is found. Additionally, Equation (Equation 29) shows that the target position of the $p_{\varepsilon }$ is the centroid of the area enclosed by the three positions obtained by observing the $p_{\alpha },\;p_{\beta },\;p_{\delta }$.

(3) Attacking Prey

During the iteration, when ψ decreases linearly from 2 to 0, its corresponding k also changes in range $\left[-\psi ,\;\psi \right]$. When the value of k is in the range, the next position of the gray wolf can be anywhere between its current position and the prey position. When |k|<1, wolves attack their prey. When |k|>1, the gray wolf separates from its prey and continues to look for more suitable prey.

The algorithm steps at time t are as follows:

(1) Initialization h, k, $x_{Ji,p}$. $x_{J_{H},\;S},\;y_{J_{H},\;S},\;z_{J_{H},\;S}$, S∈{A,B} are following uniform distribution in range [−0.2, 0.2]. $\varphi_{B}$, $\varphi_{C}$ are following uniform distribution in range $\left[0,\;\pi \right]$. $\theta_{B}$, $\theta_{C}$ are following uniform distribution in range $\left[0,\;2\pi \right]$.

(2) Calculate the individual fitness of the population by substituting $x_{J_{H},p}$ into Equation (Equation 3) and $x_{J_{K},p}$ into Equation (Equation 6). Select the three individuals with the smallest error as $p_{\alpha },\;p_{\beta },\;p_{\delta }$.

(3) Calculate the position vectors of $p_{\varepsilon }$ relative to $p_{\alpha },\;p_{\beta },\;p_{\delta }$ in γ-th iteration, respectively by Equation (Equation 27). Update the position of $p_{\varepsilon }$ by Equations (Equation 28) and (Equation 29).

(4) If the maximum number of iterations is reached, go to step 6. Otherwise, go to step 5.

(5) Reorder to determine the position of the gray wolf, and go to step 2.

(6) Output current optimal solution by Equation (Equation 29).

## 4. Calculation of Human Lower Limbs Joint Angles

To calculate the joint angle, the coordinate system of each limb needs to be constructed first. The previous method of establishing the limb coordinate system is to let the subject stand in a standard standing posture. The limitations of this method have been analyzed in Section 1. According to the calibration algorithm in Section 3, we can obtain the installation position and direction information of human lower limb sensors and use this information to establish the coordinate system attached to a limb.

### 4.1. Establish the Coordinate System Attached to a Limb

In human kinematic analysis, it is crucial to determine the spatial relationship between two adjacent limbs. The establishment of the spatial relationship between two limbs depends on the coordinate frame fixed on each limb, i.e., the coordinate system attached to a limb. The commonly used method in domestic and overseas is to establish the coordinate frame on the axis of the proximal or posterior joints of each limb [28]. In this work, according to the standards of the international society of biomechanics (ISB) [24], we establish the coordinate system attached to a limb on the proximal joint, and all the coordinate systems are the right-hand Cartesian coordinates. As shown in Figure 4, the limbs of the pelvis, upper leg, lower leg, and foot of the left leg are denoted as the rod $L_{1}$, $L_{2}$, $L_{3}$, and $L_{4}$, respectively, and the coordinate system attached to each limb is denoted as $\left\{L_{1}\right\}$, $\left\{L_{2}\right\}$, $\left\{L_{3}\right\}$, $\left\{L_{4}\right\}$. In most human motions, the pelvis generally makes only translational motion without rotation; therefore, in the case of only calculating the joint angles, the $L_{1}$ rod can be considered as a fixed rod.

As shown in Figure 5a, $\left\{L_{2}\right\}$ is established at $J_{H}$. Make the normal line of $r_{J_{K}}$ through $J_{H}$, and the foot point is N. The y-axes direction of the attached coordinate system is from N to $J_{H}$. The z-axes parallel to the knee joint rotation axis and the direction is left leg to right leg. The x-axes are perpendicular to the y- and z-axes.

In Figure 5b, the vertical line of the knee joint is made through the point $J_{A}$, and the vertical point is O, $\left\{L_{3}\right\}$ is established at O. The y-axes direction is from $J_{A}$ to O. The z-axes coincides with the knee joint axis, and the direction is the left leg to right leg. The x-axes are perpendicular to the y and z-axes, $\left\{L_{4}\right\}$ is established at $J_{A}$. At the initial stage, $\left\{L_{1}\right\}$ and $\left\{L_{2}\right\}$ coincide, and $\left\{L_{3}\right\}$ and $\left\{L_{4}\right\}$ are parallel.

The attitude transformation between two adjacent sensors is defined by Equation (Equation 30).
(30)$$\begin{array}{cc} R_{B}^{A}\left(t\right) & ={\left(R_{A}^{O_{g}}\left(t\right)\right)}^{T}R_{B}^{O_{g}}\left(t\right),\;\\ R_{C}^{B}\left(t\right) & ={\left(R_{B}^{O_{g}}\left(t\right)\right)}^{T}R_{C}^{O_{g}}\left(t\right),\\ R_{D}^{C}\left(t\right) & ={\left(R_{C}^{O_{g}}\left(t\right)\right)}^{T}R_{D}^{O_{g}}\left(t\right), \end{array}$$
where $R_{B}^{A}$ is the attitude transformation matrix between A and B. The attitude transformation matrix between B and C is denoted as $R_{C}^{B}$. The attitude transformation matrix between C and D is denoted as $R_{D}^{C}$. The attitude transformation matrix of each IMU relative to the global coordinate system at time t is denoted as $R_{S}^{O_{g}}$, S∈{A,B,C,D}. It is worth noting that only the sensor coordinate system $O_{s}$ is time-dependent, resulting in $R_{S}^{O_{g}}$ changing with time and $O_{g}$ is not time-dependent. We analyze the fusion process in detail in Section 4.3. When the human body has not started motion capture and stands still, the attitude transformations between IMUs are denoted by $R_{B}^{A}\left(0\right)$, $R_{C}^{B}\left(0\right)$, $R_{D}^{C}\left(0\right)$.

According to the analysis in Section 3, the $V_{J_{H},\;B}$, $V_{J_{K},\;B}$, $V_{J_{K},\;C}$, $V_{J_{A},\;C}$, $V_{J_{A},\;D}$, $r_{J_{K},\;B}$, and $r_{J_{K},\;C}$ can be obtained. As shown in Figure 5a,b, $V_{BN}$ and $V_{CO}$ need to be calculated to obtain the coordinate system attached to each limb. Establishing the equation of the knee joint axis by Equation (Equation 31).
(31)$$\frac{x+x_{2}}{r_{x}}=\frac{y+y_{2}}{r_{y}}=\frac{z+z_{2}}{r_{z}},$$
where ${\left[x_{2},\;y_{2},\;z_{2}\right]}^{T}$ denote the coordinates of $V_{J_{H},\;B}$ and ${\left[x_{3},\;y_{3},\;z_{3}\right]}^{T}$ denote the coordinates of $V_{J_{K},\;B}$. ${\left[r_{x},\;r_{y},\;r_{z}\right]}^{T}$ denote the coordinates of $r_{J_{K},\;B}$. ${\left[x,\;y,\;z\right]}^{T}$ is any point on a line, and point N is on this line. Assume that the coordinates of $V_{BN}$ are ${\left[a,\;b,\;c\right]}^{T}$, the equation of knee joint axis by Equation (Equation 32).
(32)$$\frac{a+x_{2}}{r_{x}}=\frac{b+y_{2}}{r_{y}}=\frac{c+z_{2}}{r_{z}}$$

The normal line of $r_{J_{K}}$ can be expressed by Equation (Equation 33).
(33)$$r_{x}\left(a+x_{3}\right)+r_{y}\left(b+y_{3}\right)+r_{z}\left(a+z_{3}\right)=0$$

According to Equations (Equation 32) and (Equation 33), $V_{BN}$ can be calculated, and $\left\{L_{2}\right\}$ is calculated by Equation (Equation 34).
(34)$$\begin{array}{cc} z^{L_{2}} & =r_{J_{K},\;B}\\ y^{L_{2}} & =\frac{V_{J_{H},\;B}+V_{BN}}{∥V_{J_{H},\;B}+V_{BN}∥}\\ x^{L_{2}} & =y^{L_{2}}\times z^{L_{2}} \end{array}$$

The attitude transformation matrix from $\left\{L_{2}\right\}$ to the coordinate system of sensor B is denoted by Equation (Equation 35).
(35)$$R_{B}^{L_{2}}={\left[x^{L_{2}},\;y^{L_{2}},\;z^{L_{2}}\right]}^{T}$$

Similarly, we can obtain $V_{CO}$, and the attitude transformation matrix from $\left\{L_{3}\right\}$ to the coordinate system of sensor C is denoted by Equation (Equation 36).
(36)$$R_{C}^{L_{3}}={\left[x^{L_{3}},\;y^{L_{3}},\;z^{L_{3}}\right]}^{T}$$

At initial stage, $\left\{L_{1}\right\}$ and $\left\{L_{2}\right\}$ coincide, and $\left\{L_{3}\right\}$ and $\left\{L_{4}\right\}$ are parallel, i.e.,
(37)$$\begin{array}{cc} R_{L_{2}}^{L_{1}} & =I,\\ R_{D}^{L_{4}} & =R_{C}^{L_{3}}R_{D}^{C}, \end{array}$$
where I is a identity matrix of 3 by 3, $R_{L_{2}}^{L_{1}}$ is a rotation matrix between $\left\{L_{1}\right\}$ and $\left\{L_{2}\right\}$, and $R_{D}^{L_{4}}$ is attitude transformation matrix from $\left\{L_{4}\right\}$ to the coordinate system of sensor D.

### 4.2. Joint Angles Calculation

According to the ISB standard [24], each joint angle of the lower limbs is the motion of the lower limbs relative to the adjacent upper limb, i.e., the upper leg is relative to the pelvis, the lower leg is relative to the upper leg and the foot is relative to the lower leg. In the 3-DOF of the joint angles, flexion/extension is β, which is the angle of rotation about the z-axes. Abduction/adduction is ϕ, which is the angle of rotation about the x-axes. Internal/external rotation is δ, which is the angle of rotation about the y-axes, according to the Z-X-Y Euler angular rotation order to calculate the joint angles. At t time, the rotation matrix of limb $L_{i}$ relative to $L_{i-1}$(i=2,3,4) can be obtained by Equation (Equation 38).
(38)$$\begin{array}{cc} R_{L_{2}}^{L_{1}}\left(t\right) & =R_{B}^{L_{2}}R_{B}^{O_{g}}\left(t\right)\\ R_{L_{3}}^{L_{2}}\left(t\right) & =R_{B}^{L_{2}}{\left(R_{B}^{O_{g}}\left(t\right)\right)}^{T}R_{C}^{B}\left(0\right)R_{C}^{O_{g}}\left(t\right){\left(R_{C}^{L_{3}}\right)}^{T},\\ R_{L_{4}}^{L_{3}}\left(t\right) & =R_{C}^{L_{3}}{\left(R_{C}^{O_{g}}\left(t\right)\right)}^{T}R_{D}^{C}\left(0\right)R_{D}^{O_{g}}\left(t\right){\left(R_{D}^{L_{4}}\right)}^{T}\;\; \end{array}$$
where $R_{L_{i}}^{L_{i-1}}$ can be expressed by Equation (Equation 39).
(39)$$\begin{array}{cc} R_{L_{i}}^{L_{i-1}} & =\left[\begin{array}{ccc} -s\delta s\beta s\phi +c\delta c\phi & -s\delta c\beta & s\delta s\beta s\phi +c\delta s\phi \\ c\delta s\beta s\phi +s\delta c\phi & c\delta s\beta & -c\delta s\beta c\phi +s\delta s\phi \\ -c\beta s\phi & s\beta & c\beta c\phi \end{array}\right]\\ & =\left[\begin{array}{ccc} r_{11} & r_{12} & r_{13}\\ r_{21} & r_{22} & r_{23}\\ r_{31} & r_{32} & r_{33} \end{array}\right], \end{array}$$
where *c* is cos and *s* is sin. Euler angle can be calculated by Equations (Equation 40)–(Equation 42).
when cosβ≠0:(40)$$\begin{array}{cc} \beta & =\mathrm{atan}2\;(r_{32},\;\sqrt{r_{12}^{2}+r_{22}^{2}})\\ \delta & =\mathrm{atan}2\;(-\frac{r_{12}}{\cos\beta },\;\frac{r_{22}}{\cos\beta }),\\ \phi & =\mathrm{atan}2\;(-\frac{r_{31}}{\cos\beta },\;\frac{r_{33}}{\cos\beta }) \end{array}$$
when $\beta =90^{\circ }$:(41)$$\begin{array}{cc} \beta & =90^{\circ }\\ \delta & =0^{\circ }\\ \phi & =\mathrm{atan}2\;(r_{13},-r_{23}) \end{array}$$
when $\beta =-90^{\circ }$:(42)$$\begin{array}{cc} \beta & =-90^{\circ }\\ \delta & =0^{\circ }\\ \phi & =\mathrm{atan}2\;(r_{13},r_{23}) \end{array}$$

### 4.3. Single IMU Attitude Fusion

To improve the accuracy of attitude acquisition by single IMU, we need to fuse the attitude rotation matrix in Equation (Equation 30). The quaternion-based attitude fusion algorithm can effectively combine the error characteristics of gyroscope and accelerometer, and improve the accuracy of attitude calculation [29]. The expression of a quaternion is defined by Equation (Equation 43).
(43)$$q=q_{0}+q_{1}i+q_{2}j+q_{3}k,$$
where *i*, *j*, *k* is an imaginary unit, $q_{0}$, $q_{1}$, $q_{2}$, $q_{3}$ is a real number, and each quaternion is a linear combination of 1, *i*, *j* and *k*.

(1) Quaternion initialization

At time t, the quaternion of attitude change is $q_{t}={\left[q_{0},\;q_{1},\;q_{2},\;q_{3}\right]}^{T}$, the attitude calculation error is $\xi_{t}={\left[\xi_{x},\;\xi_{y},\;\xi_{z}\right]}^{T}$. At the initial stage, $q_{0}$ and $\xi_{0}$ are defined by Equation (Equation 44).
(44)$$\begin{array}{cc} q_{0} & ={\left[1,\;0,\;0,\;0\right]}^{T},\\ \xi_{0} & ={\left[0,\;0,\;0\right]}^{T}, \end{array}$$

(2) Correction of angular velocity error

Based on the definition of cosine matrix and Euler angles in [30], the gravity vector of the global coordinate system can be rotated to the sensor coordinate system by Equation (Equation 45).
(45)$$\begin{array}{c} \begin{array}{cc} g_{t}^{O_{s}} & =R_{O_{s}}^{O_{g}}\left(t\right)\cdot g_{t}^{O_{g}}\\ \;\;\;\; & =\left[\begin{array}{ccc} q_{0}^{2}+q_{1}^{2}-q_{2}^{2}-q_{3}^{2} & 2(q_{1}q_{2}-q_{3}q_{0}) & 2(q_{1}q_{3}+q_{0}q_{2})\\ 2(q_{1}q_{2}+q_{0}q_{3}) & q_{0}^{2}-q_{1}^{2}+q_{2}^{2}-q_{3}^{2} & 2(q_{2}q_{3}-q_{0}q_{1})\\ 2(q_{1}q_{3}-q_{0}q_{2}) & 2(q_{2}q_{3}+q_{0}q_{1}) & q_{0}^{2}-q_{1}^{2}-q_{2}^{2}+q_{3}^{2} \end{array}\right]\left[\begin{array}{c} 0\\ 0\\ 1 \end{array}\right], \end{array} \end{array}$$
where the rotation matrix of $O_{s}$ relative to $O_{g}$ by quaternion is denoted as $R_{O_{s}}^{O_{g}}\left(t\right)$. The acceleration of gravity is denoted as $g_{t}^{O_{g}}$. The acceleration of gravity after rotation from $O_{g}$ to $O_{s}$ is denoted as $g_{t}^{O_{s}}={\left[g_{x}^{O_{s}},\;g_{y}^{O_{s}},\;g_{z}^{O_{s}}\right]}^{T}$. Equation (Equation 45) is simplified by Equation (Equation 46).
(46)$$\begin{array}{cc} g_{x}^{O_{s}} & =2(q_{1}q_{3}-q_{0}q_{2}),\\ g_{y}^{O_{s}} & =2(q_{2}q_{3}+q_{0}q_{1}),\\ g_{z}^{O_{s}} & =q_{0}^{2}-q_{1}^{2}-q_{2}^{2}+q_{3}^{2}. \end{array}$$

In the process of IMU attitude rotation, the gravity vector measured by the accelerometer is $a_{t}^{O_{s}}={\left[a_{x}^{O_{s}},\;a_{y}^{O_{s}},\;a_{z}^{O_{s}}\right]}^{T}$, and the gravity vector calculated by the attitude integrated by the gyroscope is $g_{t}^{O_{s}}={\left[g_{x}^{O_{s}},\;g_{y}^{O_{s}},\;g_{z}^{O_{s}}\right]}^{T}$. The error vector between them is $\xi_{t}={\left[\xi_{x},\;\xi_{y},\;\xi_{z}\right]}^{T}$, which is the error between the attitude integrated by the gyroscope and the attitude measured by the accelerometer. It can be expressed by cross product, and $\xi_{t}$ is defined by Equation (Equation 47).
(47)$$\begin{array}{cc} \xi_{x} & =a_{y}^{O_{s}}\cdot g_{z}^{O_{s}}-a_{z}^{O_{s}}\cdot a_{y}^{O_{s}},\\ \xi_{y} & =a_{z}^{O_{s}}\cdot g_{x}^{O_{s}}-a_{x}^{O_{s}}\cdot a_{z}^{O_{s}},\\ \xi_{z} & =a_{x}^{O_{s}}\cdot g_{y}^{O_{s}}-a_{y}^{O_{s}}\cdot a_{x}^{O_{s}}. \end{array}$$

(3) Data fusion

The cross product error is adjusted by proportional-integral (PI) controller [31] to correct the bias of the gyroscope. By adjusting the two parameters $\lambda_{p}$ and $\lambda_{i}$, the speed of the accelerometer to correct the integral attitude of the gyroscope can be controlled, where, $\lambda_{p}$ is the proportional adjustment coefficient, which is used to control the speed of the error converges to the accelerometer. Once there is a deviation in the system, the proportional adjustment will immediately produce an adjustment effect to reduce the error. $\lambda_{i}$ is the integral adjustment coefficient, which is used to control the convergence speed of gyro bias, so as to eliminate the steady-state error and improve the accuracy of the system.

(1) At time t, integrate the cross product error by Equation (Equation 48).
(48)$$\begin{array}{cc} \xi_{x}^{+} & =\xi_{x}^{-}+\xi_{x}\cdot \lambda_{i}\cdot \frac{1}{2}\Delta t,\\ \xi_{y}^{+} & =\xi_{y}^{-}+\xi_{y}\cdot \lambda_{i}\cdot \frac{1}{2}\Delta t,\\ \xi_{z}^{+} & =\xi_{z}^{-}+\xi_{z}\cdot \lambda_{i}\cdot \frac{1}{2}\Delta t, \end{array}$$

(2) The fused gyro measurements are defined by Equation (Equation 49).
(49)$$\begin{array}{cc} w_{x}^{O_{s}^{+}} & =w_{x}^{O_{s}^{-}}+\lambda_{p}\cdot \xi_{x}+\xi_{x}^{+},\\ w_{y}^{O_{s}^{+}} & =w_{y}^{O_{s}^{-}}+\lambda_{p}\cdot \xi_{y}+\xi_{x}^{+},\\ w_{z}^{O_{s}^{+}} & =w_{z}^{O_{s}^{-}}+\lambda_{p}\cdot \xi_{z}+\xi_{x}^{+}, \end{array}$$
where $\lambda_{p}=2\Delta t$. The sampling frequency is denoted as Δt. The angular velocity measured by gyroscopes in IMUs coordinate system are denoted as $w_{t}^{O_{s}}={\left[w_{x}^{O_{s}},\;w_{y}^{O_{s}},\;w_{z}^{O_{s}}\right]}^{T}$. “-” is prior-estimted, and “+” is post-estimted. Since the parameter value of PI controller needs to be dynamically adjusted according to different experimental requirements, in this work, the value of $\lambda_{i}$ cannot be too large. As shown in Figure 6, after several experimental parameter adjustments, when $\lambda_{i}$ is greater than 0.2Δt, the ξ defined by Equation (Equation 48) increases gradually. Therefore, any value less than 0.2 is acceptable, and we set $\lambda_{i}=0.1\Delta t$.

(4) Update quaternion
(50)$$\left[\begin{array}{c} q_{0}\left(t+▵t\right)\\ q_{1}\left(t+▵t\right)\\ q_{2}\left(t+▵t\right)\\ q_{3}\left(t+▵t\right) \end{array}\right]=\left[\begin{array}{c} q_{0}\left(t\right)\\ q_{1}\left(t\right)\\ q_{2}\left(t\right)\\ q_{3}\left(t\right) \end{array}\right]\left[\begin{array}{cccc} 1 & -\frac{w_{x}^{O_{s}}}{2}\Delta t & -\frac{w_{y}^{O_{s}}}{2}\Delta t & -\frac{w_{z}^{O_{s}}}{2}\Delta t\\ \frac{w_{x}^{O_{s}}}{2}\Delta t & 1 & \frac{w_{z}^{O_{s}}}{2}\Delta t & -\frac{w_{y}^{O_{s}}}{2}\Delta t\\ \frac{w_{y}^{O_{s}}}{2}\Delta t & -\frac{w_{z}^{O_{s}}}{2}\Delta t & 1 & \frac{w_{x}^{O_{s}}}{2}\Delta t\\ \frac{w_{z}^{O_{s}}}{2}\Delta t & \frac{w_{y}^{O_{s}}}{2}\Delta t & -\frac{w_{x}^{O_{s}}}{2}\Delta t & 1 \end{array}\right]$$

(5) Convert the updated quaternions into matrix forms by $R_{O_{s}}^{O_{g}}$ of Equation (Equation 45), and the attitude transformation matrix of each IMU relative to the global coordinate system can be obtained.

## 5. Experimental Analysis

### 5.1. Measurement Equipment

In Figure 7, four IMUs (Yost Labs, USA) with red marked points were bound to the limb with medical tape. Each IMU included a tri-axial accelerometer in the range ±16 g and a tri-axial gyroscope in range ±2000 deg/s. The four IMUs are bound to the waist, upper leg, lower leg, and foot, respectively, and the IMUs are connected by a 3.5 mm retractable spring cable. All IMUs record data synchronously during motion capture, and the sampling frequency is 100 Hz, which is transmitted to the computer wirelessly. The white marked point is a wireless motion capture device named Perception Neuron Pro (Noitom, CN), which is used as a reference system and worn on the limbs simultaneously with IMUs for synchronous capture. The motion capture device is only used to verify the performance of the calibration algorithm.

Considering that different gait characteristics of different people may affect the experimental results and avoid the randomness of the experimental results, in the data acquisition phase, the data of the accelerometer and gyroscope are acquired by three subjects, including one female at a height of 165 cm (subject 1) and two males at heights of 175 cm (subject 2) and 180 cm (subject 3), respectively. Among them, the data of motion capture device are only used for reference, which is different from the data collected by IMUs and they do not affect each other. As shown in Figure 8, the IMUs are mounted at position 1 and position 2, respectively, for two experiments, and the data of the accelerometer and gyroscope are collected at two different positions, respectively. To study the randomness and accuracy of the calibration algorithm, the binding positions of IMUs do not coincide with the motion capture device, i.e., the positions of IMUs is displaced. Before the formal motion capture, the subjects need to stand still for 10 s to obtain the attitude rotation matrix between adjacent IMUs in the initial state. During motion capture, the subjects walked for 15 s. To avoid the deviation of the walking mode from normal, the subjects are not informed that the walking data would be used for calibration. The data collected by IMUs were substituted into the calibration algorithm to calculate the angle changes of the hip, knee, and ankle during walking and compared with the reference value of motion capture equipment.

### 5.2. Data Analysis

To analyze the accuracy of the algorithm, we compared the deviation between the three algorithms and the reference value. The angle calculated by the GN, DWPSO, and GWO are estimated values, and the angle calculated by motion capture equipment are reference values. The Root Mean Squared Error (RMSE) between the estimate values and the reference value for each DOF is calculated by Equation (Equation 51).
(51)$$RMSE=\sqrt{\frac{1}{t}\sum_{m=1}^{t}{\left(H_{IMU,m}-H_{MCS,m}\right)}^{2}},$$
where $H_{IMU}$ is the estimation values from IMUs, $H_{MCS}$ is the angle reference value calculated by motion capture equipment named Perception Neuron Pro.

### 5.3. Results and Analysis

Figure 9, Figure 10 and Figure 11 show the RMSE comparison of three algorithms when IMUs on three subjects were bound in two positions. It shows that the three algorithms achieve position calibration at two positions, respectively. No matter where it is, the RMSE of the DWPSO is the lowest of the three algorithms, which is closer to the reference value. It is worth mentioning that the reference value will change when IMUs change the position. When the IMUs were placed in the second position, the accuracy of the three algorithms were ranked the same. When the GWO is used for position calibration, the initial population is easy to be unevenly distributed and lacks global communication, resulting in the final solution being easy to fall into local optimization. In the DWPSO algorithm, we introduce dynamic weight to control the speed of the initial population and improve the accuracy of the algorithm. Therefore, the calibration performance of the GWO is lower than DWPSO. However, the introduction of dynamic weight increases the complexity of the PSO algorithm and reduces the efficiency of DWPSO.

Table 1 shows the average and standard deviation (SD) of 15 computation times of three algorithms, and all algorithms are completed on the same computer. As shown in Table 1, the GWO uses the shortest average computation times, followed by the DWPSO, and the GN takes the longest. When a high calibration accuracy and fast algorithm efficiency are required, the GWO can be used for calibration. However, the SD value of the GWO is the highest, indicating that the algorithm is less stable than DWPSO and GN, which may reduce the efficiency. The DWPSO algorithm is relatively stable, and the optimization performance is better than the other two algorithms. When there is no requirement for speed, the DWPSO may be the best choice.

Combined with the analyses in Table 2 and Figure 9, Figure 10 and Figure 11, although the heights and sexes of the subjects are different, the variation range of the results of each subject is roughly the same, and the performance of the calibration algorithm is also the same. This is because the three calibration algorithms are carried out under the same joint constraints and the joint constraints of each subject are the same, which will not be affected by the different gait characteristics of the subjects. Therefore, subject 1 is selected as the sample for analysis. Figure 12 shows the variation of the joint angle of IMUs in position 1 for 5 s. It shows that the angle variation waveform of each joint is consistent with the reference value, only the up and down translation is produced in terms of amplitude. It indicates that the offset error of IMUs position is fixed and will not change over time.

Table 2 shows the test results of the RMSE when IMUs on three subjects were bound in position 1. In $H_{\mathrm{FE}}$, $H_{\mathrm{IE}}$, $K_{\mathrm{FE}}$ and $A_{\mathrm{IE}}$, the performances of the three algorithms are close to each other. In $H_{\mathrm{AA}}$, $K_{\mathrm{AA}}$, $K_{\mathrm{IE}}$, $A_{\mathrm{FE}}$ and $A_{\mathrm{AA}}$, the calibration performances of the DWPSO and GWO are better than GN. A possible explanation is that under this DOF, the variation of the joint is not significant, which will affect the calculation of the Jacobian matrix and the accuracy of the calibration. The DWPSO and GWO do not consider the Jacobian matrix, and their accuracy is significantly higher than GN. Additionally, the results in Figure 12 show that when the joint angle is around $0^{\circ }$, the values of the DWPSO and GWO are closer to the reference value than GN, e.g., $K_{\mathrm{AA}}$ or $A_{\mathrm{FE}}$. This is because the DOF is not the main activity of the joint, and will also affect the performance of the GN algorithm.

To more intuitively evaluate the consistency between the results of the three calibration algorithms and the reference, we selected the angle values of $H_{\mathrm{AA}}$ and $K_{\mathrm{IE}}$ in Figure 12 as samples and plot the Bland–Altman diagram for analysis. As shown in Figure 13, the x-axes are the average of each individual between the reference value and estimated value, the y-axes are the difference of each individual between the reference and the estimated. The two red lines in the figure are the upper and lower limits of the 95% consistency interval, the purple dotted line indicates that the average value of the difference is 0, and the green line is the average value of the difference between the reference value and the estimated value in each individual. The closer the green line is to the purple dotted line, the higher the consistency between the reference value and the estimated value. As shown in Figure 13a, in $H_{\mathrm{AA}}$, the average difference value of the DWPSO is the closest to 0, and the consistency with the reference value is the highest, the GWO is the second, and the GN is the lowest. As shown in Figure 13b, the consistency analysis of $K_{\mathrm{IE}}$ is also the highest in the DWPSO. These results are consistent with the curve results in Figure 12. Additionally, most of the results in Figure 13 are within the confidence interval, which explains why the waveforms of the estimated value and the reference value are similar in Figure 12.

Through the above analysis, it shows that in the IMUs position calibration of three joints of human lower limbs, the three algorithms have achieved good calibration results, and the calibration accuracy of the population algorithm is better than GN. When the joint changes sufficiently in a certain DOF, the results of the three algorithms are close. When the joint changes are insufficient, the calibration accuracy of the population algorithm is obviously better than GN. For two different population algorithms the DWPSO and GWO, different choices can be made according to practical applications.

## 6. Conclusions

In this work, we introduce the DWPSO, GWO, and GN algorithms to realize the dynamic calibration of IMUs’ positions based on human lower limb joint constraints. The performance of the algorithm is evaluated by gait experiments. The results show that the three algorithms have achieved IMU position calibration and are suitable for estimating the angles of the hip, knee, and ankle of humans during free walking. The simulation results show that the DWPSO has the best calibration performance, followed by the GWO and GN. When the joint rotation is sufficient or the joint is in the main motion, the performances of the three algorithms are close. When the joint rotation is insufficient, the performances of the DWPSO and the GWO are significantly better than the GN.

At present, our work has achieved an IMU position calibration of human lower limbs. However, when applied for a whole-body calibration, a large amount of data may cause the decline of the searchability of the DWPSO and GWO. In future work, we need to conduct further experiments.

Another route of future work is that when the offset error of IMUs position drifts slowly over time in the short term, an accelerometer and gyroscope can be combined to estimate the joint axis of the knee joint, and further improve the position calibration accuracy.

## Figures and Tables

**Figure 1 sensors-21-07161-f001:**
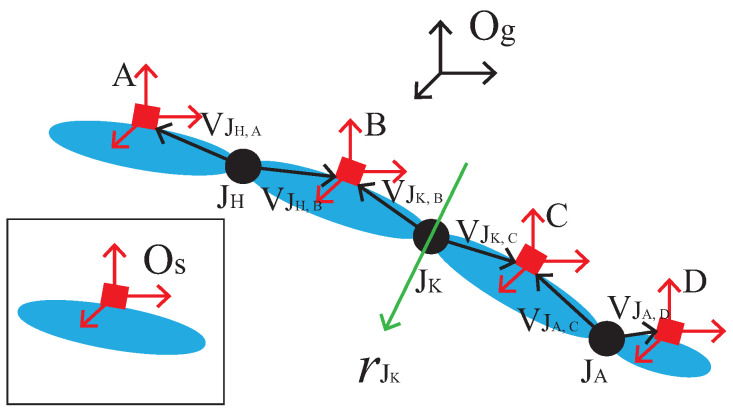
A model of a human left lower limb connected by joints.

**Figure 2 sensors-21-07161-f002:**
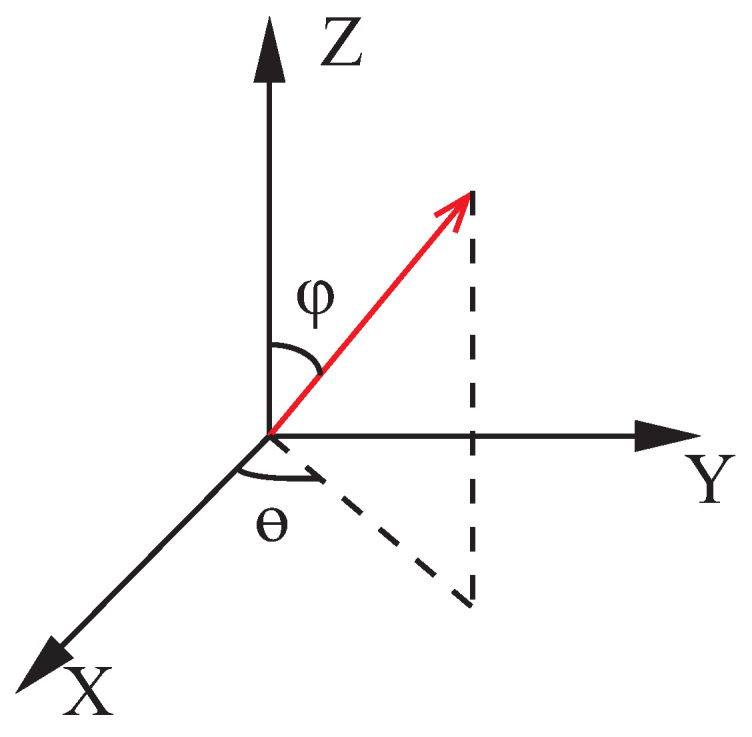
Spherical coordinates of $r_{J_{K},\;B}$ and $r_{J_{K},\;C}$.

**Figure 3 sensors-21-07161-f003:**
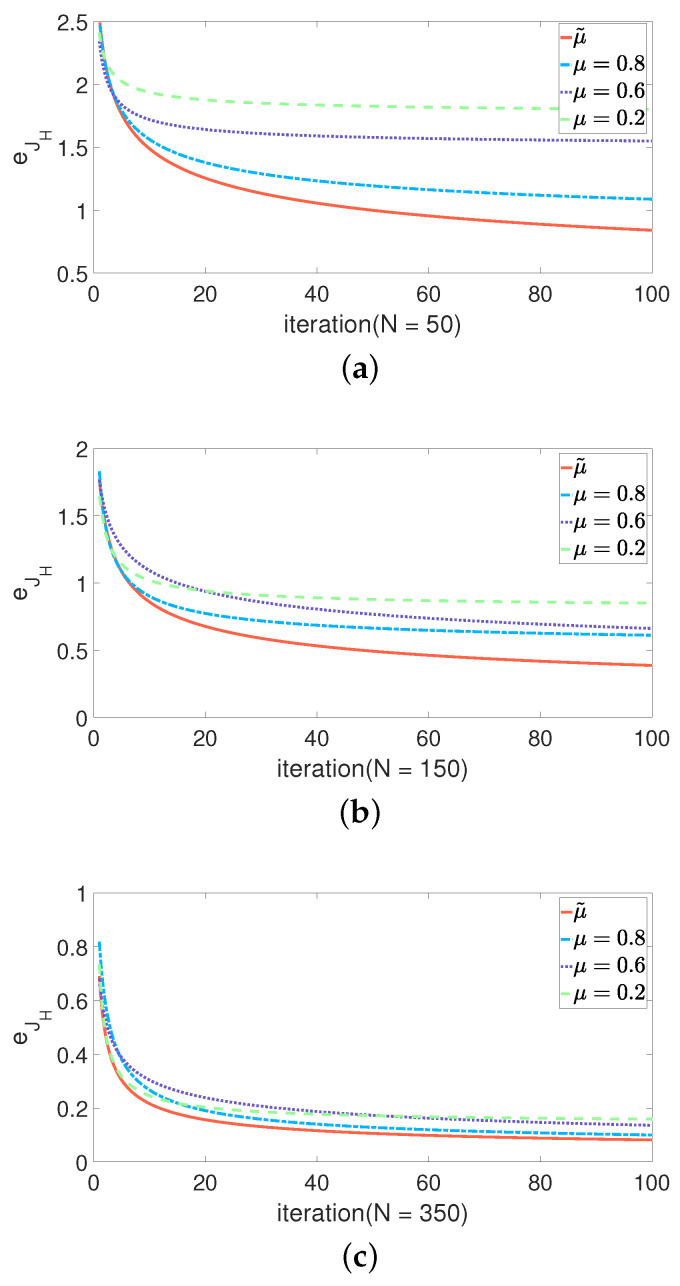
Convergence curves of dynamic weight and fixed weight with different numbers of particles. (**a**) N=50; (**b**) N=150; (**c**) N=350.

**Figure 4 sensors-21-07161-f004:**
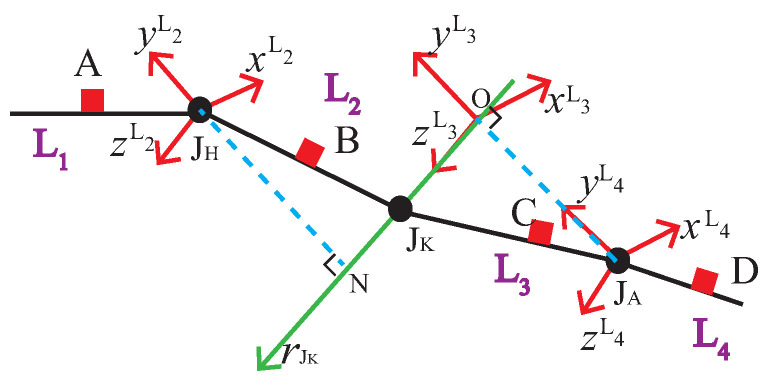
Structural diagram of human lower limb.

**Figure 5 sensors-21-07161-f005:**
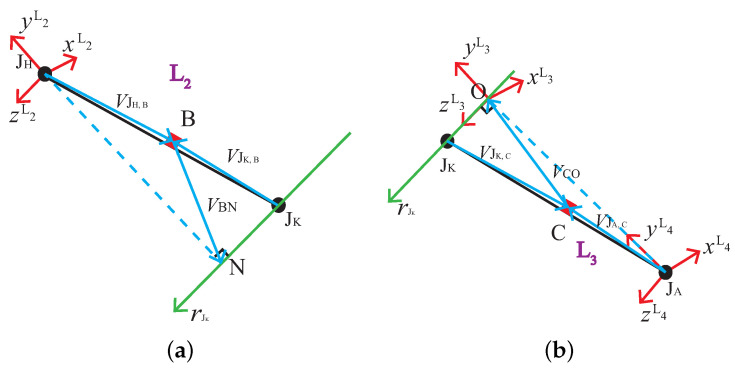
Structure diagram of limbs and position vector of each joint. (**a**) Limb $L_{2}$; (**b**) Limb $L_{3}$.

**Figure 6 sensors-21-07161-f006:**
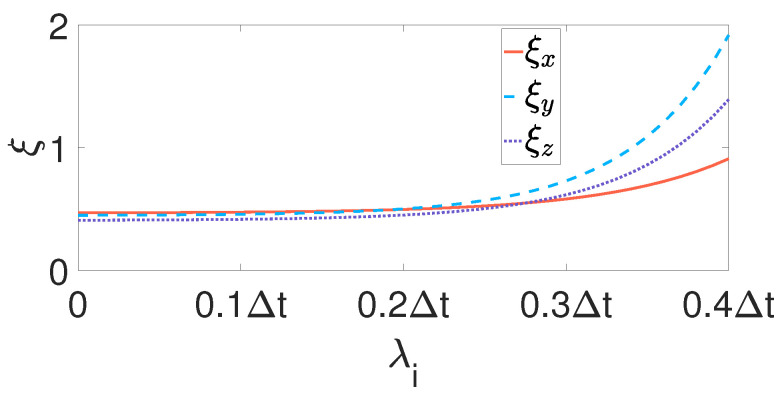
When the $\lambda_{i}$ takes different values, the convergence curve of ξ.

**Figure 7 sensors-21-07161-f007:**
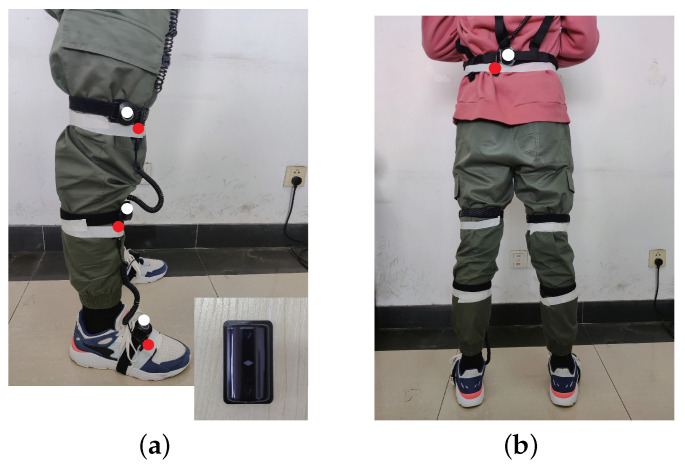
The motion capture test equipment. (**a**) he IMUs on the leg; (**b**) the IMUs on the waist.

**Figure 8 sensors-21-07161-f008:**
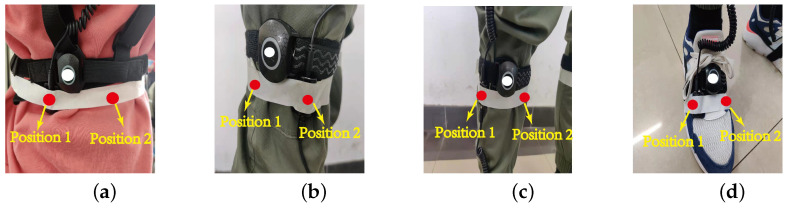
The IMUs are mounted in position 1 and position 2 for the experiment, respectively. (**a**) The waist; (**b**) the upper leg; (**c**) the lower leg; (**d**) the foot.

**Figure 9 sensors-21-07161-f009:**
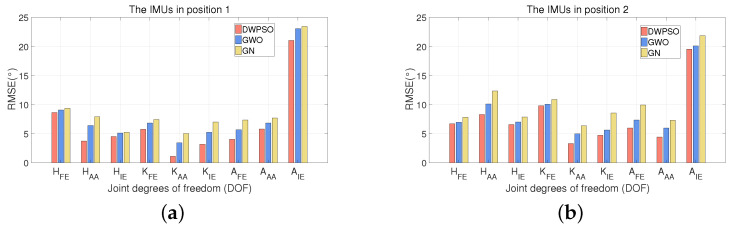
The RMSE (${}^{\circ }$) comparison of three algorithms when IMUs on subject 1 were bound in two positions. (**a**) The IMUs in position 1; (**b**) the IMUs in position 2.

**Figure 10 sensors-21-07161-f010:**
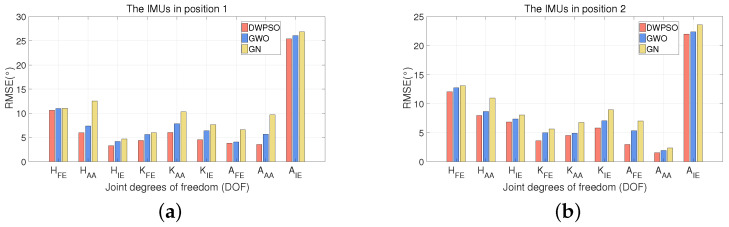
The RMSE (${}^{\circ }$) comparison of three algorithms when IMUs on subject 2 were bound in two positions. (**a**) The IMUs in position 1; (**b**) the IMUs in position 2.

**Figure 11 sensors-21-07161-f011:**
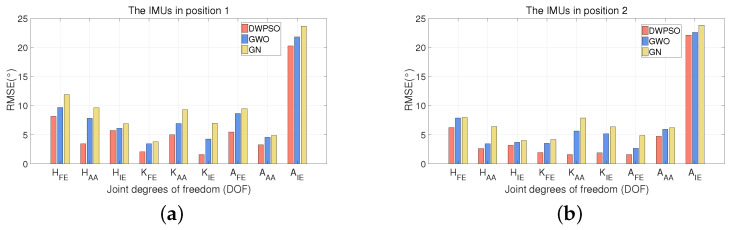
The RMSE (${}^{\circ }$) comparison of three algorithms when IMUs on subject 3 were bound in two positions. (**a**) The IMUs in position 1; (**b**) the IMUs in position 2.

**Figure 12 sensors-21-07161-f012:**
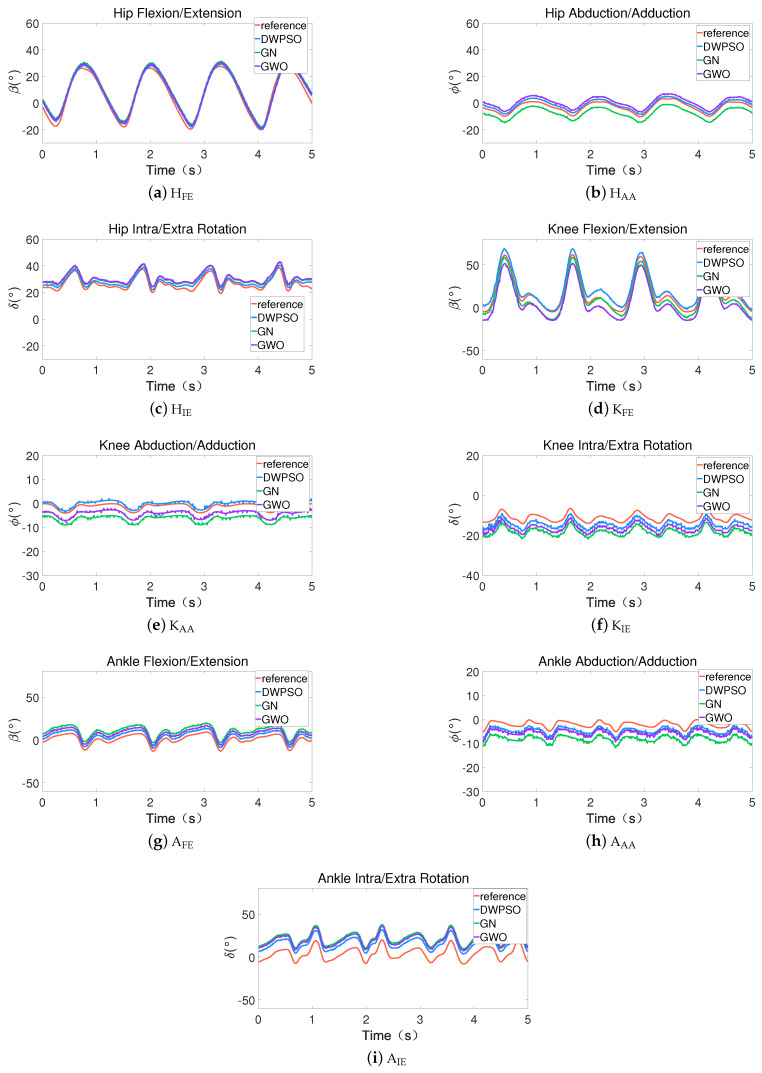
The variation of the joint DOF when IMUs on subject 1 were bound in position 1. (**a**–**c**) The hip joint; (**d**–**f**) the knee joint; (**g**–**i**) the ankle joint.

**Figure 13 sensors-21-07161-f013:**
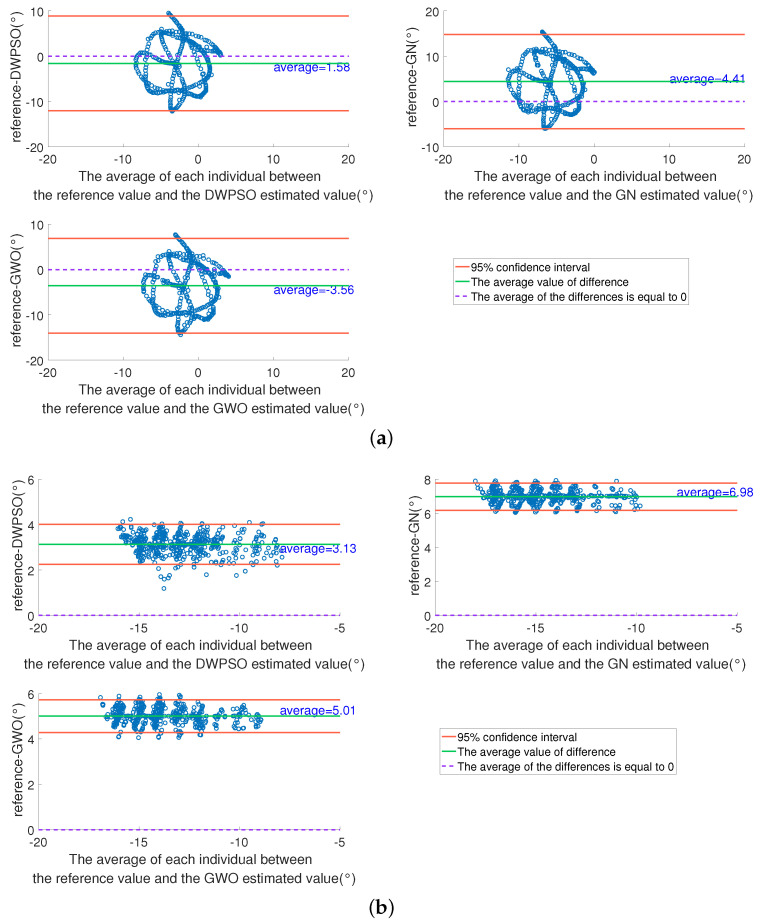
The Bland-Altman consistency analysis of estimated and reference values of three algorithms when IMUs in position 1. (**a**) $H_{\mathrm{AA}}$; (**b**) $K_{\mathrm{IE}}$.

**Table 1 sensors-21-07161-t001:** Average and standard deviation (SD) of 15 computation times of the DWPSO, GWO, and GN.

Algorithm Type	Average (s)	SD
DWPSO	1076.1	2.01
GWO	576.3	3.76
GN	1556.4	2.98

**Table 2 sensors-21-07161-t002:** The RMSE (${}^{\circ }$) comparison of three algorithms when IMUs on three subjects were bound in position 1.

	Subject 1			Subject 2			Subject 3		
	**DWPSO**	**GWO**	**GN**	**DWPSO**	**GWO**	**GN**	**DWPSO**	**GWO**	**GN**
$H_{\mathrm{FE}}$	8.65	9.09	9.36	10.63	10.97	11.05	8.17	9.65	11.90
$H_{\mathrm{AA}}$	3.72	6.42	7.90	5.97	7.36	12.53	3.42	7.83	9.61
$H_{\mathrm{IE}}$	4.53	5.13	5.26	3.29	4.15	4.69	5.71	6.08	6.86
$K_{\mathrm{FE}}$	5.77	6.86	7.45	4.35	5.61	5.99	2.06	3.41	3.76
$K_{\mathrm{AA}}$	1.12	3.42	5.02	6.01	7.86	10.34	4.98	6.93	9.28
$K_{\mathrm{IE}}$	3.16	5.26	7.02	4.54	6.37	7.63	1.57	4.25	6.97
$A_{\mathrm{FE}}$	4.03	5.69	7.36	3.81	4.08	6.62	5.43	8.62	9.45
$A_{\mathrm{AA}}$	5.76	6.83	7.71	3.55	5.67	9.71	3.26	4.54	4.86
$A_{\mathrm{IE}}$	21.05	23.07	23.45	25.41	26.06	26.83	20.25	21.79	23.67

## Data Availability

All measurement data in this paper have been listed in the content of the article, which can be used by all peers for related research.

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
