# Peer review of "Joint Constraints Based Dynamic Calibration of IMU Position on Lower Limbs in IMU-MoCap"

_sensors, 2021, doi:10.3390/s21217161_

Round 1

Reviewer 1 Report

In this resubmission, the authors have responded appropriately to most of my previous comments. I think, most notably is the addition of two more test subjects and one additional placement of the IMUs. The new results better support the conclusions and the authors have improved their discussion of limitations and future work.

There are however, several old issues that still exist and some issues with the new additions. Most pressing is the overall quality of the presentation and writing. There are numerous grammatical errors and typos throughout the paper that makes it difficult to read. In particular, the new additions to the introduction needs to be proofread and improved. I highly recommend that the authors employ proffessional English proofreading.

There are also several issues with the technical notation and descriptions of the algorithms. Here I will provide detailed comments:

  1. Equation 15 (and others): Time dependence should not appear in superscript.
  2.  Grey wolf optimizer:
    1. Citations are missing for this method.
    2. Equation 27: Are the f-quantities calculated for every individual epsilon? If so f should be functions of epsilon.
    3. Equation 28: What are subscripts p_1, p_2, p_3? They are not explained. Should this equation not show the update of every epsilon's position?
    4. Equations 27&28: What is the reasoning behind using two random variables h and k instead of just one? Both h and k operate on the f-variables.
    5. Line 272: How is individual fitness calculated?
    6. Line 274: It does not seem correct that eq. 27 calculates the distance between grey wolves epsilon and the alpha, beta and gamma wolves, because of the random variables h that operate on the positions. It seems to calculate how closely epsilon will follow alpha, beta, gamma when their positions are updated, which is subject to some random factor.
  3. Line 294: "anterior joint" should be "proximal joint".
  4. Eq 30 and other places: Time dependence of the rotation matrices should not be placed in the superscript. It can be interpreted then that it's the reference frame A or O_g that is time-dependent, when it really is the rotation between two moving coordinate systems, the matrix R itself, that is time dependent. In addition, O_g should not be time-dependent if fixed and constant with respect to the local environment.
  5. Eq 40, 41, 42: You have replaced the previous function f with atan2 in eq 40. But not in 41 and 42.
  6. The attitude fusion algorithm:
    1. Section 4.3 (3) Data fusion. It seems that the attitude estimator used is a complementary filter, which can work very similarly to a PI controller. Since we are dealing with an estimation problem, it would be more appropriate to consider the complementary filter since this algorithm has been used in many previous studies for attitude estimation.
    2. Eq. 44: "int" looks like it does not belong here. Also appears in subsequent equations.
    3. Eq. 48: Should say /Delta t instead of just t?

I also have some comments addressing the authors' response letter regarding my previous comments:

Response 1: Main reason for including the accelerometer is not to handle a drifting error. It is because it allows the knee joint axis to be identified even if the knee angles rotate very little. Furthermore, using the accelerometer also removes the sign-ambiguity of the joint axis, which comes from the fact that the gyroscope constraint (Eq. 6) will be satisfied regardless if r_{J_K} have positive or negative sign. The joint axis in frame B will be in the same global direction as the joint axis in frame C. 

Reviewer 2 Report

Review of sensors 1407679

After reading the paper, I do not feel enough of a subject matter expert in the algorithmic design sections to give a fair critique and review to those sections. I concentrated on the remaining sections in my review.

All references are page and line numbers from the September 19, 2021 version pdf.

15 – “joint variation obviously” reword

16 – I do not understand this sentence. Possibly remove “both”, remove “position” or “displacement”, remove “reality” 

25 – Change to “IMU-based“

217 - Where does the -0.2 to 0.2 range come from?

402 – Capitalize “perception neuron pro” properly

404 – How were the devices synchronized? Are all the devices streaming to the same software or do they use their own data capture utility?

434 – remove “both”

437 – why does the reference value change between positions 1 and 2?

439 – From the figures, it looks like GN consistently has the highest RMSE then GWO then DWPSO. In this line, you seem to claim that the DWPSO and GWO are higher than GN. Do you mean something like “GN is the highest” rather than “are higher than GN” here?

Table 1 – Maybe both the average and standard deviation would be good here to get a sense of a constant time speed up between algorithms

Table 2 -- Are these the data from Figs. 9a-11a? They seem to not agree with the plots

Figure 12 – Maybe put the labels that you use in the text and tables (e.g., HFE and HAA) in each subfigure to compare the text more easily to the figure?

Overall

You are using solely RMSE to compare the accuracies of the three optimizations to the “true” reference angles. Looking at Fig. 12 though, I think that you may be underselling the accuracies of the three algorithms? It looks like the main reason for the “high” RMSE values is a vertical offset from the reference rather than a high deviation in the reported values. Perhaps a Bland-Altman plot would give better insight into the reliability of the algorithms? Perhaps reporting the ICC would provide a better quantitative comparison between the methods? Is there a way to correct the offset error in the algorithms?  

I think that you need more quantifiable statistical evidence to support your conclusions about the performance of the algorithm. You are currently relying on visual inspection of the difference in RMSE values between only three subjects each with two markers on different limbs. You need a more holistic statistical analysis of the difference in quality between the algorithms.

Round 2

Reviewer 1 Report

Most of my comments regarding technical errors and errors in notation have been resolved. Although the authors claim to have had the English writing proofread, there are still plenty of typos and grammatical errors. I will list the ones that stood out to me. I urge the authors to be more thorough with internally reviewing their manuscripts in the future as it appears careless and ignorant when such errors persists through multiple revisions.

Line numbers are from the revised manuscript with highlighted changes. I could not find the clean revised manuscript.

Line 56: "rapid rotate the joint" change to "fast rotations of the joint"

Line 60: Capitalization on "work". Further comment: Introducing most references with "Study in .." and "Work in .." appears repetetive.

Line 269, 270, 275, 280 (among others): Consider not beginning sentences with variables, use instead e.g. "The positions of ... in the /gamma-th iteration are denoted by ..."

Line 271: Wrong grammar, change "is the position vector difference .." to "are the position vectors relative to ...".

Line 300: Remove " : ", change to ".. population by substituting .. into Eq. 3..".¨

Line 301: Change "select three individuals" to "select the three individuals".

Line 347: Here you say initial values are at t=0, previously, t=1 was used (eq. 13).

Line 389: To say that "int \xi_t" is the initial value updated at each time appears strange. The initial value should only be the value at t=0, to say that it is updated in each time step and still call it initial is confusing and seems wrong. I suggest removing the "int" notation completely as it can also be confused with integral or integer.

Line 469-470: Would be more correct to say "When the IMUs were placed in the second position, the accuracy of the three algorithms were ranked the same".

Figures 9-11: Better to say that the sensors are in positions 1 and 2 instead of subjects, since it is actually the sensor placement that is different. Should be changed throughout the manuscript.

Line 480: Change "calculation time" to "computation time".

Table 1: Change caption to "Average and standard deviation (SD) of 15 computation times ...".

Table 2: Typo, "Subjet" should be "Subject".

Figure 13: I do not understand these Bland-Altman plots. What does "Mean of the reference and the [Method]" mean?. Also there are no units on the axes.

Author Response

This manuscript is a resubmission of an earlier submission. The following is a list of the peer review reports and author responses from that submission.

Round 1

Reviewer 1 Report

The topic of IMU-based joint angle estimation is important for ambulatory human movement analysis. The authors try to exploring constraints to improve the estimation accuracy, which is a good idea.

However, this paper is not well organized. The are several major issues that need the author’s attention. The introduction did not clearly state What is unknown and What is known. “Study in [7] has xxx” is not a typical way on introducing a study. In the method section, it is unclear that “3) Data fusion Line 242”. In the experiments, only one healthy subject was recruited, which is not sufficient to demonstrate the advantage of the proposed algorithm. “named perception neuron pro of model Noitom, CN,” is not a standard practice. Normally, the company name will be included in brackets. For “5.3. Results and Analysis”, the discussion needs further improvement to show the importance of the proposed method. The conclusion section needs to be more concise. Comparison with previous work should be in Discussion Section.

Reviewer 2 Report

The authors proposed a dynamic calibration of IMUs position Jacobian matrix and particle swarm optimization (PSO) algorithm. 

The comments are listed as follows.

1. The introduction part seems too long but not a logical presentation.   This section is recommended that the author refines with containing the scope, the significance of the research, and highlighting the potential outcomes.

2.The results (section 5) should be further elaborated to show how they could be used for real applications. Recommend authors compare the results of the proposed approach with a variety of genetic algorithms (GA) and PSO.

3. The comparison is restricted to the angle variation of the cases. It would be nice if the author(s) attempt(s) to compare the proposed scheme with other population-based meta-heuristic algorithms of different genres.  
4. Future research direction needs to be strengthened.

Reviewer 3 Report

This paper aims to evaluate the performance of sensor-to-segment calibration methods for IMUs when applied to joint angle estimations of lower limbs. The main contribution is the evaluation on a human subject with a reference motion capture system. The authors compare the use of two different optimization methods; Gauss-Newton (GN) and Particle Swarm Optimization (PSO). It is found that PSO performs better than GN for joints where the walking motion do not fully excite all degrees of freedom.

The presentation of the paper in terms of quality of writing and use of technical notation is lacking. It is strongly suggested that the authors employ a third party to thoroughly review the English writing of their manuscript. For the technical issues please find my comments below:

Major comments:

1) Equation 4: Some recent studies have shown that including accelerometer information in the calculation of the joint axis will result in a more accurate calibration. See e.g. F. Olsson et al. "Robust Plug-and-Play Joint Axis Estimation Using Inertial Sensors", Sensors, 20(12), 2020. and D. Nowka et al. "On Motions That Allow for Identification of Hinge Joint Axes from Kinematic Constraints and 6D IMU Data", 18th European Control Conference (ECC), 2019. Have the authors considered this?

These references are also recommended since they describe further development of the method proposed by T. Seel et al. in 2012, which is cited by the authors as [18].

2) The error plots in Figure 3 are not clearly defined. What error is shown here and what is the cost function that is being minimized? Is real IMU data included?

3) Equation 31: Is this update rule really valid for the gyroscope data? It appears as if two different quantities are added together, w - the gyroscope data which has a unit of rad/s and xi - the attitude error between the estimated gravity vector and the acceleromter which has a unit of m/s^2 or g. The result of this update should not be able to yield a quantity of unit rad/s as the left hand side suggests.

Also, what is the purpose of lambda in Eq. 31 and why is the value of 0.005\Delta t chosen? Is the term with lambda really neccessary? It appears that xi^- (superscript minus) is also included in xi^+ (through Eq. 30) so the lambda term in 31 could be avoided by adapting h in Eq. 30?

Until this can be clarified by the authors, Eq. 31 does not appear to be technically correct.

4) Equation 31: Should it be w^{O_s +} or w^{O_s -} here? Also w^{O_s} on line 246 should be in bold font to be consistent with previous notation?

5) Figure 5: Since the model assumes that the knee joint is a hinge joint (1-DOF joint), that means that the bar connecting the joint centres (J_H to J_K and J_K to J_A) should form a right angle with the joint axis r_{J_K}. After reading the section following Figure 5, it appears that the points N and O correspond to the true knee joint center rather than J_K. 

Also, the coordinate systems depicted in Figure 5, do not match their definitions in the text. The y-axes should point from N to J_H and from J_A to O.

6) The z-axes from the coordinate systems 2-4 are all assumed to be parallell with the joint axis r_{J_K}, is this a reasonable assumptions?

7) Lines 302-304: According to the definitions of your coordinate systems in Figure 5 and the following section, flexion/extension is about the z-axes, abduction/adduction is about the x-axes and internal/external rotation is about the y-axes.

8) Lines 325-327: You claim here that placing the IMUs at different locations than the reference motion capture system allows you to study the "randomness and accuracay of the calibration algorithm". Should not multiple different IMU placements be tested to better study the performance of the calibration method.

Here, only one subject and one trial of 5s walking is used to evaluate the methods. This does not seem to be adequate data to support the claims the authors make in their conclusions.

9) Lines 329-330: Is the same 5s of walking data used to both calibrate and assess the performance of the methods? Standard practice would be to use separate calibration and testing data.

10) Section 5.2 Data Analysis: Here, the authors compare the joint angles after the two calibration methods are used to the reference motions capture system and joint angles obtained from un-calibrated IMUs (UNCal). It is not clear to me how joint angles can be obtained without any type of calibration since the coordinate systems still have to be defined. How are the coordinate systems defined in the UNCal setting?

11) Conclusions: The authors do not discuss shortcomings and limitations of their study. I do not think that it is possible to make such a strong claim with the limited amount of data used in the study (5s of walking data from one subject with sensors placed in only one position). The authors should add a discussion regarding limitations and prospects for future research.

Minor comments:

A) Line 230-231: It is not correct to say that the gravity vector (3 dim) is estimated by a quaternion (4 dim).

B) Equation 27: The rotation matrix is also time dependent, does also apply to other equations with rotation matrices of Os relative to Og.

C) Equation 37-40: (and in other places): Using integer digits in superscript (like R^2 and x^2) can be confused with a power, I suggest changing this notation throughout the manuscript.

D) Line 310: Why not simply put atan2 instead of f in the equations?

E) Line 358: Better to say "reference value" than "real value" since the measurements of the reference motion capture system also contains a degree of uncertainty.

F) Tables 1-3: Should contain the units of the RMSE.